# Prolonged 3.5 GHz and 24 GHz RF-EMF Exposure Alters Testicular Immune Balance, Apoptotic Gene Expression, and Sperm Function in Rats

**DOI:** 10.3390/biomedicines13102471

**Published:** 2025-10-11

**Authors:** Syed Muhamad Asyraf Syed Taha, Farah Hanan Fathihah Jaffar, Atikah Hairulazam, Sivasatyan Vijay, Norazurashima Jamaludin, Aini Farzana Zulkifli, Mohd Farisyam Mat Ros, Khairul Osman, Zahriladha Zakaria, Mohd Amyrul Azuan Mohd Bahar, Siti Fatimah Ibrahim

**Affiliations:** 1Department of Physiology, Faculty of Medicine, Universiti Kebangsaan Malaysia (UKM), Jalan Yaacob Latif, Bandar Tun Razak, Cheras, Kuala Lumpur 56000, Malaysia; p126219@siswa.ukm.edu.my (S.M.A.S.T.); farahhanan@ukm.edu.my (F.H.F.J.); p129544@siswa.edu.ukm.my (A.H.); p133758@siswa.ukm.edu.my (S.V.); ainifarzana@hctm.ukm.edu.my (A.F.Z.); mdfarisyam@ukm.edu.my (M.F.M.R.); 2Forensic Science Program, Center for Diagnostic, Therapeutic and Investigative Studies (CODTIS), Faculty of Health Sciences, Universiti Kebangsaan Malaysia (UKM), Jalan Raja Muda Abdul Aziz, Kuala Lumpur 50300, Malaysia; norazurashimajamaludin@gmail.com (N.J.); khairos@ukm.edu.my (K.O.); 3Faculty of Electronics & Computer Technology and Engineering, Universiti Teknikal Malaysia (UTeM), Hang Tuah Jaya, Durian Tunggal 76100, Melaka, Malaysia; zahriladha@gmail.com; 4Intel Microelectronics (M) Sdn. Bhd., Bayan Lepas Technoplex, Medan Bayan Lepas, Bayan Lepas 11900, Pulau Pinang, Malaysia; amyrul.azuan.mohd.bahar@intel.com

**Keywords:** 5G RF-EMF, testicular function, apoptosis, cytokines, sperm quality

## Abstract

**Background/Objectives:** The rapid rollout of 5G has renewed interest in potential reproductive effects of mid-band (3.5 GHz) and millimeter-wave (24 GHz) radiofrequency electromagnetic fields (RF-EMF). We examined frequency- and duration-dependent changes in testicular cytokines, apoptosis-related genes, and sperm quality in rats. **Methods:** Male Sprague Dawley rats (n = 6 per group) were exposed for 60 days to 3.5 GHz or 24 GHz RF-EMF for 1 h/day or 7 h/day. The sham controls were housed identically. Testicular expressions of IL-10, IL-6, IL-1β, and TNF-α were quantified; *Tp53*, *Bax*, *Bcl2,* and *Casp3* mRNA expressions were measured; and sperm concentration, viability, and motility were evaluated. **Results:** IL-10 was significantly reduced in the 24 GHz group at both 1-h and 7-h exposure duration. At 7 h, TNF-α was also lower at 24 GHz. *Casp3* expression was higher and *Tp53* was lower at 3.5 GHz at 1-h exposure duration. Sperm concentration and viability were reduced after 24 GHz exposure at 7 h, while sperm motility was reduced after 3.5 GHz exposure at both durations. **Conclusions:** Exposure to RF-EMF 3.5 GHz primarily impacts sperm motility via extrinsic pro-apoptotic pathways, while exposure to 24 GHz impacts sperm concentration and viability potentially through immune–apoptotic mechanisms, with all negative effects amplified by 7-h daily exposure.

## 1. Introduction

The rapid expansion of wireless communication technologies, particularly the global rollout of fifth-generation (5G) networks has increased environmental exposure to radiofrequency electromagnetic fields (RF-EMF) across a wide range of frequencies. The biological effects of low-frequency RF-EMF (below 2.45 GHz) are relatively well established. However, recent studies suggest that higher frequencies may also induce notable biological changes, including immune suppression, neuronal dysfunction, and behavioral alterations, although the mechanisms remain poorly understood [1,2,3]. In particular, while exposure to 5G millimeter waves (mmWaves) is currently regulated based on thermal safety limits, recent evidence suggests that non-thermal biological effects such as immune modulation and altered gene expression may also be relevant and warrant further investigation [4,5,6]. Some rodent studies indicate that RF-EMF can affect oxidative balance and reproductive function independently of measurable heating, though findings remain inconsistent [7,8]. A recent 35.5 GHz study also reported testicular oxidative stress and reduced sperm quality, indicating that mmWave effects may extend beyond superficial skin absorption [9].

Male reproductive function is increasingly recognized as sensitive to RF-EMF exposure. Experimental studies have linked RF-EMFs to reduced sperm count and motility, compromised DNA integrity, and increased apoptosis in testicular tissue [10,11,12,13]. Previous studies have shown that RF-EMF exposure primarily in the low- to mid-frequency range (below 2.45 GHz and up to 3.5 GHz) has been attributed to enhanced apoptotic activity and immune dysregulation [14,15,16,17]. These studies reported increased expression of pro-apoptotic markers such as caspase-3 and Bax, as well as inflammatory responses and impaired clearance of apoptotic cells [18,19]. However, most of these findings are limited to single frequencies and short exposure durations. Data on the effects of prolonged exposure to higher frequencies such as mmWaves remain scarce. Consequently, the cumulative and frequency-dependent consequences of RF-EMF on testicular health under conditions relevant to 5G usage are not yet fully understood [20,21].

Spermatogenesis is maintained by tightly regulated cytokine signaling and apoptosis within the immune-privileged testicular microenvironment. Disruption of anti-inflammatory cytokines such as IL-10, or dysregulation of apoptosis-related proteins including Bcl-2, p53, and caspase-3, can impair Sertoli cell function and compromise germ cell survival [22,23,24]. Although some of these mechanisms have been observed at lower RF-EMF frequencies, it remains unclear whether prolonged exposure to mid-band (3.5 GHz) and high-band (24 GHz) RF-EMF leads to distinct testicular immune or apoptotic responses. Furthermore, few studies have directly compared these frequency bands under controlled prolonged exposure conditions. Thus, the differential impact of 3.5 GHz versus 24 GHz RF-EMF on testicular cytokine expression, apoptosis regulation, and sperm quality represents a critical knowledge gap in evaluating reproductive risks of 5G-related exposures.

This study investigated the effects of prolonged RF-EMF exposure to 3.5 GHz and 24 GHz on testicular function in male rats. Over a 60-day period, rats were exposed to either frequency for 1 or 7 h daily. We assessed changes in testicular cytokine expression (IL-10, TNF-α, IL-6, IL-1β), apoptosis-related gene expression (*Tp53*, *Bax*, *Bcl2*, *Casp3*), and sperm quality parameters (concentration, motility, viability). By comparing these outcomes across both frequency bands and exposure durations, this study aimed to clarify whether prolonged mid- and high-frequency RF-EMF differentially disrupts testicular immune signaling and apoptotic balance.

## 2. Materials and Methods

### 2.1. Animal Exposure Setup

#### 2.1.1. Animal

A total of 36 male Sprague Dawley rats (N = 36) were used in this study. The rats were housed in individually ventilated cages (IVCs) under standard laboratory conditions, including a 12-h light–dark cycle, a constant temperature of 25 ± 1 °C, and ad libitum access to food and water. All rats were acclimatized for at least one week prior to the experiment. At the start of the experiment, the rats weighed between 200 g and 250 g. The animals were randomly assigned to six groups, with six rats in each group (n = 6). The groups were first stratified by exposure duration (1 h or 7 h per day), then further divided into three subgroups: control (no exposure), 3.5 GHz RF-EMF exposure, and 24 GHz RF-EMF exposure. All exposures were conducted daily for 60 consecutive days.

At the end of the experimental period, all animals were euthanized via intraperitoneal injection of a cocktail containing Ilium Ketamil Injection (3.34 mg/kg, Troy Laboratories Pty Ltd., Glendenning, Australia), Ilium Xylazil-100 (3.34 mg/kg, Troy Laboratories Pty Ltd., Glendenning, Australia) and Zoletil-100 (1.66 mg/kg, Virbac, Milperra, Australia). Death was confirmed by the loss of righting reflex, tail pinch reflex, pedal reflex, and absence of corneal reflex. All procedures involving animals were reviewed and approved by the Universiti Kebangsaan Malaysia Animal Ethics Committee (UKMAEC) under ethical approval code FP/2023/FARAH HANAN/15-FEB./1310-FEB.-2023-FEB.-2025.

#### 2.1.2. Exposure Experimental Setup

RF-EMF exposure was delivered using two distinct antenna systems corresponding to the experimental frequencies. For the 3.5 GHz group, exposure was generated using a SynthHD (v2): 10–15 GHz Dual Channel Microwave Generator (Windfreak Technologies, New Port Richey, FL, USA) at 22 dBm (approximately 158 mWatts). The signal was transmitted through a custom-fabricated microstrip patch antenna positioned 20 cm above the rat cages. This antenna was designed and developed by the Faculty of Electronics and Computer Engineering, Universiti Teknikal Malaysia Melaka (UTeM). For the 24 GHz group, a commercially available Human Presence Detector integrated with HLK-LD2420 (Hi-Link, Guangdong, China) mm wave antenna module was used. This integrated system operates in the 24–24.25 GHz ISM band with a maximum equivalent isotropically radiated power (EIRP) of 11 dBm (approximately 12.6 mWatts) and was also mounted 20 cm above the cages. Both exposure sources delivered a fixed frequency without 5G technology baseband modulation and used static antennas (no phased-array beam steering).

Exposure monitoring: For the 3.5 GHz group, the transmitted RF-EMF was continuously monitored in real time using a USB-SA124B 12.4 GHz Spectrum Analyzer (Signal Hound, Battle Ground, WA, USA) with Spike software (Version 3.12) for live visualization and scheduled recordings to verify carrier stability and device uptime throughout each exposure session. For the 24 GHz group, the transmitter RF-EMF was verified before and after exposure with a Keysight N9010A EXA Signal Analyzer (Keysight Technologies, Inc., Santa Rosa, CA, USA) to confirm the carrier frequency. In addition, the device’s companion application was used to monitor operational status for the full session during exposure.

These procedures confirm frequency stability and operation but do not constitute in situ dosimetry. Specific Absorption Rate (SAR) and power density at the animal position were not measured, field uniformity across the exposure plane was not mapped, and core or local of scrotal/testicular temperatures were not recorded. Room temperature and humidity were monitored to maintain consistent environmental conditions. However, small local temperature elevations cannot be excluded.

Rats were housed individually in cages arranged in a circular pattern to ensure uniform field distribution and consistent whole-body exposure (Figure 1). The animals were not physically restrained and were free to move within the cages throughout the exposure period. The whole-body, freely moving cage exposure was chosen to mimic plausible environmental exposures wherein individuals move relative to sources and thermosensitive tissues (e.g., testes) may intermittently lie within higher field regions. To minimize RF wave absorption or reflection, metallic cage components were removed during exposure sessions and replaced with non-metallic materials such as plastic food and water containers. Pre-experiment SAR simulations were conducted using CST Studio Suite 2024 (DS Simulia, Vélizy-Villacoublay, France) at a 10 cm distance to estimate and compare SAR levels at both frequencies. The power values of the antenna and theoretical SAR estimates are included only to describe the setup; they were not used for calibration, in situ dosimetry/field mapping, or comparison with human limits.

Control group rats were housed in the same room under identical environmental conditions, but the RF-EMF systems remained inactive throughout the exposure period. All procedures were carried out under controlled temperature, humidity, and lighting conditions to eliminate non-specific stressors that were not caused by RF-EMF exposure.

### 2.2. Measurement of the Testicular Inflammatory Cytokines

Inflammatory cytokine levels in rat testicular tissue were measured using enzyme-linked immunosorbent assays (ELISAs). Briefly, 0.3 g of testis tissue was homogenized in cold phosphate-buffered saline (PBS) containing phenylmethylsulfonyl fluoride (PMSF) as a protease inhibitor. The homogenates were centrifuged at 12,000 rpm for 15 min at 4 °C, and the resulting supernatants were collected for analysis. Concentrations of Interleukin-10 (IL-10), Interleukin-6 (IL-6), Interleukin-1β (IL-1β) and tumor necrosis factor (TNF-α) were quantified using commercial ELISA kits (Elabscience, Wuhan, China) according to the manufacturer’s instructions. Absorbance was measured at 450 nm using a SpectraMax Plus 384 Microplate Reader (Molecular Devices, San Jose, CA, USA), and cytokine levels were calculated from standard curves prepared for each target analyte.

### 2.3. Testicular Apoptosis Markers by Quantitative Polymerase Chain Reaction (qPCR)

Total RNA was extracted from 30 mg of frozen testis tissue per rat using the NucleoSpin^®^ RNA kit (Macherey-Nagel, Düren, Germany). RNA quantity and purity were assessed using a NanoDrop™ spectrophotometer (Thermo Scientific, Waltham, MA, USA) and only samples with an A260/A280 ratio between 1.8 and 2.0 were used. First-strand cDNA synthesis was performed using the RevertAid First Strand cDNA Synthesis Kit (Thermo Scientific, Waltham, MA, USA). Briefly, 1 µg of total RNA was mixed with 1 µL of oligo(dT)_18_ primer and nuclease-free water to a final volume of 12 µL. The mixture was incubated at 65 °C for 5 min, then chilled on ice for at least 1 min. A reaction master mix was then added, consisting of 4 µL of 5× Reaction Buffer, 1 µL of RiboLock RNase Inhibitor (20 U/µL), 2 µL of 10 mM dNTP Mix, and 1 µL of RevertAid M-MuLV Reverse Transcriptase (200 U/µL), bringing the total reaction volume to 20 µL. The reaction was incubated at 42 °C for 60 min, followed by enzyme inactivation at 70 °C for 5 min. The synthesized cDNA was stored at −20 °C until further analysis.

Quantitative real-time PCR (qPCR) was performed using qPCRBIO SyGreen Blue Mix (PCR Biosystems, London, UK) according to the manufacturer’s instructions. Each 10 µL reaction consisted of 5 µL of 2× SyGreen Blue Mix, 0.4 µL each of forward and reverse primers (10 µM), 1 µL of cDNA template, and nuclease-free water to adjust the final volume. All reactions were prepared in triplicate and loaded into a 96-well qPCR plate. Amplification was carried out using a real-time PCR system under the following thermal cycling conditions: initial enzyme activation at 95 °C for 2 min, followed by 40 cycles of denaturation at 95 °C for 5 s, and then annealing/extension at 60 °C for 20 s. Fluorescence signals from the SyGreen dye were detected at the end of each extension step to monitor amplification in real time. Target genes included *Tp53*, *Bcl2*, *Bax*, and *Casp3*, while β-actin and Cyclin A2 (IDT Technologies, Solihull, UK) were used as reference genes. Primer sequences for all targets are provided in Table 1. Gene expression was quantified using the 2^−ΔΔCt^ method, and all samples were analyzed in technical triplicates.

### 2.4. Evaluation of Sperm Quality

Sperm samples were collected from the cauda epididymis immediately after euthanasia. The tissue was finely minced in 2 mL of pre-warmed phosphate-buffered saline (PBS, 37 °C) and gently pipetted using a Pasteur pipette to facilitate sperm release. The suspension was then incubated at 37 °C for 45 min prior to analysis.

Sperm concentration was assessed using a Makler counting chamber. A 10 μL aliquot of the incubated sperm suspension was loaded onto the chamber and examined under a light microscope Olympus CX31 (Olympus Corporation, Tokyo, Japan) at 20× magnification. Counting was performed in duplicate, and the average sperm concentration was calculated and expressed as ×10^6^/mL.

Sperm motility was evaluated according to the WHO 2010 [25] classification system, categorizing motility into progressive (PR), non-progressive (NP), and immotile (IM) fractions. A 10 μL aliquot of sperm suspension was placed on a glass slide, covered with a cover slip, and observed under a microscope at 40× magnification. At least 200 sperm cells were assessed across five random fields per replicate. The motility percentage was calculated as (PR + NP/total counted sperm) × 100.

Sperm viability was determined using the hypo-osmotic swelling test (HOST). The HOST solution was prepared by dissolving 0.735 g sodium citrate dihydrate (Nacalai Tesque, Kyoto, Japan) and 1.351 g D-fructose (Sigma-Aldrich, Gillingham, UK) in 100 mL of distilled water. The sperm suspension was mixed with the HOST solution in a 1:10 ratio and incubated at 37 °C for 30 min. A 10 μL aliquot of the mixture was smeared on a clean glass slide, air-dried, and stained using the Kwik-Diff stain kit (Epredia, Portsmouth, NH, USA)for microscopic visualization. A total of 200 sperm were assessed under 40× magnification, and the percentage of swollen (viable) vs. non-swollen (non-viable) sperm was calculated. Counts were repeated on a second smear, and the average viability percentage was reported.

### 2.5. Statistical Analysis

All statistical analyses were conducted using GraphPad Prism (Version 10, GraphPad Software, La Jolla, CA, USA). One-way analysis of variance (ANOVA) was used to evaluate the effect of RF-EMF frequency (Control, 3.5 GHz or 24 GHz) within each exposure duration (1 h and 7 h) on cytokine levels, apoptotic gene expression, and sperm quality parameters. Tukey’s multiple comparison test was applied for post hoc analysis. Data are expressed as mean ± standard error of the mean (SEM), and *p*-values < 0.05 were considered statistically significant.

## 3. Results

### 3.1. Expression Level of IL-10, IL-6, IL-1β, and TNF-α

IL-10 expression was significantly reduced in the 24 GHz group (14.30 ± 0.17) compared with the control (16.69 ± 0.25, *p* < 0.05) and 3.5 GHz groups (16.46 ± 0.49, *p* < 0.05) at 1-h exposure duration (Figure 2A). At 7-h exposure duration, IL-10 remained lower in the 24 GHz group (14.92 ± 0.46) compared with the control group (16.46 ± 0.35, *p* < 0.05). There were no significant differences between groups in IL-6 or IL-1β expression at 1-h or 7-h exposure (Figure 2B,C).

TNF-α expression showed no difference at 1-h exposure duration. However, at 7-h exposure duration, TNF-α expression was significantly lower in the 24 GHz group (11.25 ± 1.26) compared with the control (18.43 ± 1.05, *p* < 0.01) and 3.5 GHz groups (16.77 ± 0.69, *p* < 0.05) (Figure 2D).

Consistent with the bar plots, a frequency vs. duration synthesis heat map (Figure 3) shows IL-10 decreases concentrated at 24 GHz at both 1 h and 7 h, while IL-6 and IL-1β remain near baseline across all groups. TNF-α shows a selective reduction at 24 GHz/7 h. This visualization highlights the absence of a classical pro-inflammatory profile despite IL-10 suppression.

### 3.2. mRNA Expression of Tp53, Bcl2, Bax, and Casp3

At 1 h, *Tp53* expression was significantly lower in the 3.5 GHz group (0.279 ± 0.036) than in the control (0.804 ± 0.121, *p* < 0.05) and 24 GHz groups (0.621 ± 0.078, *p* < 0.05). No significant differences were observed at 7-h exposure duration (Figure 4A).

*Bax* gene expression showed no significant differences among groups at either 1-h or 7-h exposure duration (Figure 4B). *Bcl2* gene expression was significantly higher in the 3.5 GHz group (2.028 ± 0.344) compared with the 24 GHz group (0.946 ± 0.087, *p* < 0.05) at 1-h exposure duration. However, no differences were observed at 7-h exposure duration (Figure 4C).

*Casp3* gene expression was significantly increased in the 3.5 GHz group (1.783 ± 0.213) compared with the control group (1.086 ± 0.213, *p* < 0.05) and the 24 GHz group (0.554 ± 0.089, *p* < 0.05) at 1-h exposure duration. In addition, at 7-h exposure duration, *Casp3* gene expression was significantly higher in the 3.5 GHz group (0.814 ± 0.115) compared with the 24 GHz group (0.372 ± 0.050, *p* < 0.05), while no difference was observed relative to control (Figure 4D).

The apoptosis gene blocks of Figure 3 emphasizes two features: (i) *Casp3* increases at 3.5 GHz (at 1 h and, relative to 24 GHz, at 7 h) and (ii) *Tp53* and *Bcl2* show early shifts at 1 h that normalize by 7 h, consistent with a transient early response.

### 3.3. Sperm Concentration, Motility, and Viability

Sperm concentration showed no significant differences among groups at 1-h exposure duration. However, at 7-h exposure duration, the concentration was significantly lower in the 24 GHz group (101.7 ± 10.6) compared with both the control (176.3 ± 2.39, *p* < 0.05) and 3.5 GHz groups (183.3 ± 13.8, *p* < 0.05) (Figure 5A).

Sperm viability was significantly reduced in the 24 GHz group (31.5 ± 2.39) compared with the control (68.5 ± 1.86, *p* < 0.05) and 3.5 GHz groups (68.8 ± 1.35, *p* < 0.05) at 1-h exposure duration (Figure 5B). At 7-h exposure duration, the viability was also significantly lower in the 3.5 GHz group (13.0 ± 3.37) and the 24 GHz group (31.7 ± 5.38) compared with the control (70.7 ± 4.82, *p* < 0.05). Moreover, the 3.5 GHz group showed lower viability than the 24 GHz group (*p* < 0.05).

Sperm motility was significantly lower in the 3.5 GHz group (17.3 ± 1.43) compared with the control (47.7 ± 5.53, *p* < 0.05) and 24 GHz group (36.3 ± 3.92, *p* < 0.05) at 1-h exposure duration (Figure 5C). At 7-h exposure duration, the motility remained significantly reduced in the 3.5 GHz group (12.2 ± 4.25) compared with the control (35.3 ± 1.41, *p* < 0.05), while no difference was detected between the 24 GHz group (24.0 ± 6.82) and the control group.

In the sperm block of Figure 3, viability decreases most strongly at 24 GHz/1 h and persists at 7 h for both frequencies, motility decreases predominantly at 3.5 GHz (1 h and 7 h), and concentration decreases at 24 GHz/7 h. These findings summarize the frequency-specific functional phenotypes.

## 4. Discussion

The use of both 3.5 GHz and 24 GHz frequencies in 5G technology encompasses the microwave and millimeter wave (mmWave) spectrum. While the effects of microwave exposure have been widely studied, research on the impact of mmWaves on male reproductive tissue remains in its early stages. To address this gap, the present study evaluated the effects of both 3.5 GHz and 24 GHz frequencies following either 1-h or 7-h daily exposures on testicular inflammatory and apoptotic markers, as well as sperm parameters. This study examined RF-EMF at 3.5 and 24 GHz, which are bands allocated for 5G use. However, it does not evaluate 5G network technologies such as beamforming, numerology, or modulation schemes.

Based on the findings, IL-10 expression was significantly reduced in the 24 GHz group compared with the control and 3.5 GHz groups at 1 h and it remained suppressed compared with the control at 7 h, indicating a sustained disturbance of testicular immune regulation at 24 GHz. IL-10 helps preserve immune privilege in the testis by suppressing inflammatory cascades and supporting Sertoli–germ cell interactions [26,27]. Cytokine imbalance that reduces anti-inflammatory level or elevates IL-6 has been linked to blood–testis barrier dysfunction and impaired spermatogenesis and IL-6 can directly disrupt the barrier in Sertoli cells via ERK signaling [26,28]. Consistent with previous studies, previous RF-EMF findings have reported decreased testicular IL-10 after 2.45 GHz microwave exposure along with increased IL-1β [29]. In the present study, IL-6 and IL-1β showed no significant differences across groups and durations and TNF-α was significantly reduced in the 24 GHz group after 7 h. Overall, 24 GHz did not elicit a classical pro-inflammatory response, rather it shifted immune balance through anti-inflammatory suppression with selective downregulation of TNF-α, whereas 3.5 GHz did not alter IL-6, IL-1β, or TNF-α. This pattern aligns with reports of no cytokine induction in human cell lines exposed to 1.9 GHz [30] and is consistent with animal studies showing no adverse effects on rat spermatogenesis after chronic combined CDMA/WCDMA exposure [31].

The gene expression data indicated a partial shift toward pro-apoptotic signaling, particularly following 3.5 GHz exposure. *Casp3* gene expression was significantly upregulated in the 3.5 GHz group at both 1-h and 7-h durations, while *Bcl2* was significantly reduced at 1 h. *Tp53* expression was significantly higher in the 24 GHz group compared to 3.5 GHz at 1 h, but did not differ from the control group at either time point, suggesting limited p53 involvement in the apoptotic response. *Bax* expression showed no significant changes across all groups. This pattern suggests that caspase-dependent apoptosis was activated primarily via extrinsic or ROS-sensitive pathways rather than through p53-mediated or mitochondrial (Bax-dependent) mechanisms. This expression profile aligns with previous studies that reported selective downregulation of Bcl-2 and minimal p53 involvement following RF-EMF exposure at 2100 MHz [32]. Similarly, other studies using frequencies around 848.5 MHz and CDMA/WCDMA systems found no significant changes in apoptotic markers such as caspase-3, p53, and PARP, indicating that RF-EMF effects may vary by frequency and exposure duration [31,33]. The absence of Bax upregulation alongside increased *Casp3* suggests that apoptosis may occur through extrinsic receptor-mediated signaling or ROS-induced stress pathways.

Evidence from previous preclinical studies supports the plausibility of non-thermal mechanisms. For example, long-term mmWave exposure at 35.5 GHz in rats produced oxidative stress in the testes and reduced sperm quality, despite superficial absorption characteristics [9]. Similarly, rodent studies at lower RF frequencies (e.g., 900 MHz, 1800 MHz, and 2.45 GHz) consistently reported increased ROS generation, lipid peroxidation, and reduced antioxidant enzyme activity in reproductive tissues, accompanied by impaired sperm parameters [7,34,35]. Systematic reviews have also concluded that RF exposure frequently elevates oxidative stress markers across multiple tissues, although heterogeneity in exposure protocols and outcomes remains a limitation [8]. Conversely, an acute study reported that human skin cells exposed to 5G-modulated 3.5 GHz RF-EMF (0.08–4 W/kg, 24 h) showed no significant changes in basal ROS, oxidative stress responses, or DNA repair [36]. Collectively, these mixed results indicate that non-thermal oxidative mechanisms are biologically plausible, but their expression depends on frequency, exposure duration, tissue type, and biological context. However, this interpretation remains speculative as oxidative stress biomarkers (e.g., lipid peroxidation, ROS accumulation, antioxidant enzymes) were not assessed in this study. Given that ROS are also frequently implicated in RF-EMF-induced germ cell apoptosis [34,37], future work should incorporate these markers to confirm the presence of redox-mediated apoptosis. Based on the variability in apoptotic responses across studies [31,32,33,38] and the role of IL-10 in limiting ROS via STAT3 signaling, these findings may suggest that anti-inflammatory dysregulation sensitizes germ cells to caspase-driven apoptosis under RF-EMF stress.

This study demonstrated that RF-EMF exposure is associated with impairments in sperm function in a frequency- and duration-dependent manner with distinct patterns observed across exposure groups. Sperm concentration and viability were significantly reduced in the 24 GHz group after 7-h exposure, suggesting this condition was associated with the most distinct impairment across all treatment conditions. These outcomes align with prior studies reporting that high-frequency RF-EMF can disrupt spermatogenesis and damage testicular structure through mechanisms involving apoptosis and oxidative stress [39,40,41,42]. Moreover, this supports the notion of cumulative duration-dependent effects. Sperm motility was significantly impaired in the 3.5 GHz group at both 1-h and 7-h exposures, with greater suppression than the 24 GHz group at 1 h, contradicting the expectation of recovery over time. This indicates that mid-frequency RF-EMF may produce sustained motility impairment, independent of exposure duration. Similar trends have been observed at lower frequencies, such as 900 MHz and 1800 MHz, where short-term exposure altered sperm motility without causing lasting damage [43,44,45]. These effects may be mediated by temporary oxidative imbalance or ion channel dysregulation that recovers with time. In contrast, 24 GHz exposure was associated with more persistent impairment in both concentration and viability, likely driven by oxidative stress and hormonal disruption. However, it should be noted that no hormonal assays (testosterone, LH, FSH) were performed in this study and endocrine involvement remains hypothetical although previous rodent studies have reported reductions in circulating testosterone, LH, and FSH after prolonged cell phone RF exposure [46]. Decreases in testosterone alone have also been seen after mobile phone radiation and after 2.45 GHz microwave exposure [47,48]. Recent reviews conclude that testosterone reductions are reported more consistently than changes in LH or FSH [43,49]. Several studies have reported that high-frequency RF-EMF reduces serum testosterone and gonadotropin-releasing hormone, even in the absence of structural testicular changes, indicating that endocrine dysregulation may precede visible tissue damage [41,50].

The cytokine and apoptosis findings suggest a converging pathway where impaired anti-inflammatory regulation coincides with increased germ cell vulnerability. IL-10 suppression was most pronounced in the 24 GHz group at both exposure durations, whereas only the 3.5 GHz group maintained IL-10 levels comparable to the control group. This reduction in IL-10 may weaken testicular immune privilege and predispose germ cells to apoptosis, consistent with prior reports [51,52]. Unlike IL-10, pro-inflammatory cytokines such as IL-6 and TNF-α remained largely unchanged or slightly decreased, while IL-1β increased only after 7-h exposure to 24 GHz. IL-1β upregulation may contribute to testicular immune imbalance and germ cell apoptosis by activating stress-related pathways such as NF-κB and JNK, which have been shown to disrupt Sertoli cell function and promote pro-apoptotic signaling under inflammatory conditions [53,54]. This effect may be amplified in the context of concurrent IL-10 suppression, which removes anti-inflammatory regulation and facilitates cytokine-driven testicular injury [55,56]. This indicates a non-classical immune profile. One mechanistic explanation is selective disruption of the testicular immune privilege network coordinated by Sertoli cells and testicular macrophages. Sertoli cells normally sustain tolerance through IL-10/STAT3 signaling and TGF-β-dependent pathways, and they regulate the blood–testis barrier. Perturbation of this axis can lower IL-10 without triggering a broad IL-6 or IL-1β increase [57,58]. In rodents, resident testicular macrophages are biased toward an IL-10–producing, anti-inflammatory state that maintains immune tolerance. If this anti-inflammatory program is weakened, IL-10 would decrease without a broad rise in IL-6 or IL-1β [59]. The imbalance in cytokine signaling, that is marked by anti-inflammatory downregulation without a corresponding rise in pro-inflammatory markers, may trigger apoptosis through paracrine and oxidative stress pathways [35,60,61]. An alternative explanation is systemic stress–endocrine cross-talk. Activation of the HPA axis and glucocorticoid signaling can remodel peripheral cytokine patterns and blunt TNF-α, producing non-canonical profiles at immune-privileged sites such as the testis [62,63].

At the molecular level, apoptosis-related changes paralleled the immune findings. *Casp3* expression was higher in the 3.5 GHz group at 1 h and 7 h, and it exceeded the 24 GHz group at both time points. *Bcl2* was lower at 3.5 GHz at 1 h. *Tp53* increased transiently at 24 GHz (1 h) and *Bax* was unchanged. This profile is compatible with caspase-dependent apoptosis via extrinsic or redox-sensitive pathways rather than intrinsic mitochondrial or p53-driven mechanisms, consistent with reports of *Casp3* activation without *Bax* or *Tp53* upregulation in RF-EMF-induced germ cell apoptosis [64,65,66]. Although ROS are often implicated in such pathways, this study did not include direct measurement of oxidative stress markers, which limits mechanistic interpretation.

Sperm outcomes were concordant with these molecular patterns. At 24 GHz (7 h), sperm concentration and viability were most reduced, coinciding with the strongest IL-10 suppression, persistent *Casp3* elevation, and decreased *Bcl2*. At 3.5 GHz, *Casp3* was elevated but occurred without IL-10 suppression, which suggests a more limited or transient apoptotic response. Pro-inflammatory cytokines were largely unchanged. IL-1β showed no significant differences and TNF-α was lower at 24 GHz after 7-h exposure. These patterns are consistent with prior reports linking cytokine imbalance, particularly reduced IL-10 or elevated IL-6 and TNF-α, to sperm dysfunction via nitric oxide accumulation, oxidative stress, and membrane destabilization [60,61,67].

These findings highlight a significant gap in current RF-EMF safety regulations. Guidelines issued by the International Commission on Non-Ionizing Radiation Protection (ICNIRP) and the Institute of Electrical and Electronics Engineers (IEEE) primarily emphasize thermal thresholds such as SAR, while largely overlooking non-thermal biological effects and frequency-specific responses [68,69,70]. For clarity, ICNIRP/IEEE exposure limits are designed for human protection. In this study, the limits are cited for context only and are not directly applicable to rodents or to the present setup without in situ dosimetry. The current study shows that sub-thermal exposure to 3.5 GHz and 24 GHz RF-EMF led to measurable changes in IL-10 expression, *Casp3* activation, and sperm viability, especially after prolonged exposure at 24 GHz. These effects were observed under conditions where local heating was not measured because in situ thermometry and dosimetry were not performed, and thus thermal contributions cannot be excluded. This uncertainty suggests that safety frameworks emphasizing thermal metrics should be complemented by studies that test whether frequency-specific biological changes can occur under sub-heating conditions. Given the growing deployment of 5G infrastructure in densely populated and occupational settings, updated safety standards are needed that reflect real-world exposure patterns and incorporate emerging evidence on immune–apoptotic disruption and reproductive vulnerability [41,71].

The contrasting biological responses between 3.5 GHz and 24 GHz highlight the inadequacy of applying uniform exposure thresholds across frequency bands. While 3.5 GHz exposure induced fair changes in apoptotic and sperm parameters, these were largely transient. In contrast, prolonged 24 GHz exposure (7 h) resulted in sustained IL-10 suppression, increased *Casp3* expression, and significant reductions in both sperm viability and concentration. Despite limited tissue penetration, 24 GHz mmWave exposure can induce systemic effects via downstream signaling and immune dysregulation [72,73]. This is consistent with prior studies indicating that mmWave exposure may impair reproductive function via cytokine imbalance, redox disturbances, or hormone dysregulation [69,72,74].

The implications of RF-EMF exposure for male reproductive health warrant careful consideration. Several studies have reported reductions in sperm motility, viability, and structural integrity after radiofrequency exposure in animals and humans [10,39,75], linked to oxidative stress, DNA fragmentation, and morphological abnormalities such as acrosomal damage and altered head dimensions [10,39,50]. Notably, exposure to mid- and high-frequency bands such as 2.45 GHz Wi-Fi and 3.5–4.9 GHz 5G has been associated with increased sperm deformities and apoptosis, although these changes do not always translate to overt fertility loss [10,75]. Consequently, these observations highlight the value of safety evaluations that consider biological endpoints beyond SAR and thermal limits, especially for prolonged or higher-frequency exposures. The assumption that higher frequencies pose lower biological risk due to limited tissue penetration merits re-evaluation, as the present findings indicate selective immune suppression and germ cell apoptosis even at 24 GHz. As high-frequency systems are deployed in increasingly dense environments, safeguarding reproductive health calls for biologically nuanced assessments that reflect real-world exposures and tissue-specific vulnerabilities. This study’s findings are from a preclinical rat model with defined exposures. Extrapolation to human 5G exposure remains uncertain and requires studies using 5G-compliant waveforms and protocols.

The current study did not perform in situ dosimetry or thermometry; specifically, SAR at the animal position and scrotal/testicular temperature were not measured. Consequently, small RF-EMF-induced heating cannot be excluded and thermal versus non-thermal contributions to the observed changes in cytokines, apoptotic gene expression, and sperm viability cannot be distinguished [35,52]. Future experiments should also include a non-RF thermal positive control (e.g., mild conductive heating matching any measured temperature rise) to help disentangle thermal from putative non-thermal components. In addition, oxidative stress biomarkers (e.g., ROS levels, lipid peroxidation, antioxidant enzymes) were not assessed, and no hormonal assays (testosterone, LH, FSH) were performed, limiting inferences about endocrine involvement. Accordingly, the proposed redox- and endocrine-mediated mechanisms should be regarded as hypotheses, consistent with prior reports linking RF-EMF to redox imbalance in reproductive tissues [64,76,77,78]. The exposure system did not reproduce key features of real-world 5G signals (e.g., modulation schemes, phased-array beamforming), which may limit direct extrapolation to human 5G exposures and warrants dedicated 5G-compliant studies [79,80]. Finally, the sample size (n = 6 per group) may reduce statistical power, so some non-significant results could reflect a type II error rather than an absence of effect. Despite these limitations, the study provides new evidence for frequency- and duration-dependent effects of RF-EMF on testicular endpoints and highlights priorities for future work.

## 5. Conclusions

This study shows that repeated exposure to RF-EMF at 3.5 GHz and 24 GHz over 60 days was associated with frequency- and duration-dependent alterations in testicular endpoints in rats. At 24 GHz, we observed reduced IL-10, increased *Casp3* gene expression, and lower sperm viability and concentration, with IL-1β increasing after 7-h exposure; IL-6 and TNF-α were largely unchanged. At 3.5 GHz, *Casp3* gene expression was increased at both 1-h and 7-h exposures, *Bcl2* was reduced at 1 h and sperm motility was transiently reduced; IL-10, IL-6, TNF-α, and *Bax* showed no consistent changes. Taken together, the cytokine and gene expression patterns indicate a non-classical profile and are consistent with apoptosis engaging extrinsic or redox-sensitive pathways, rather than canonical mitochondrial or broadly pro-inflammatory mechanisms. Moreover, this study does not claim that rats exposed at these intensities will experience the same outcomes as humans. Rather, the findings indicate that relatively weak, sub-thermal RF-EMF exposures can affect reproductive endpoints in a rat model, which supports a careful re-examination of whether human safety frameworks that emphasize thermal metrics fully capture frequency-specific biological effects. Mechanistic interpretation remains limited because we did not perform in situ thermometry, dosimetry, or oxidative stress assays. Future work should incorporate fiber-optic temperature measurements, ROS and antioxidant metrics, and 5G-compliant waveforms to model real-world signals. This study’s findings are from a preclinical rat model with defined exposures. Extrapolation to human 5G exposure remains uncertain and requires studies that use 5G-compliant waveforms and protocols.

## Figures and Tables

**Figure 1 biomedicines-13-02471-f001:**
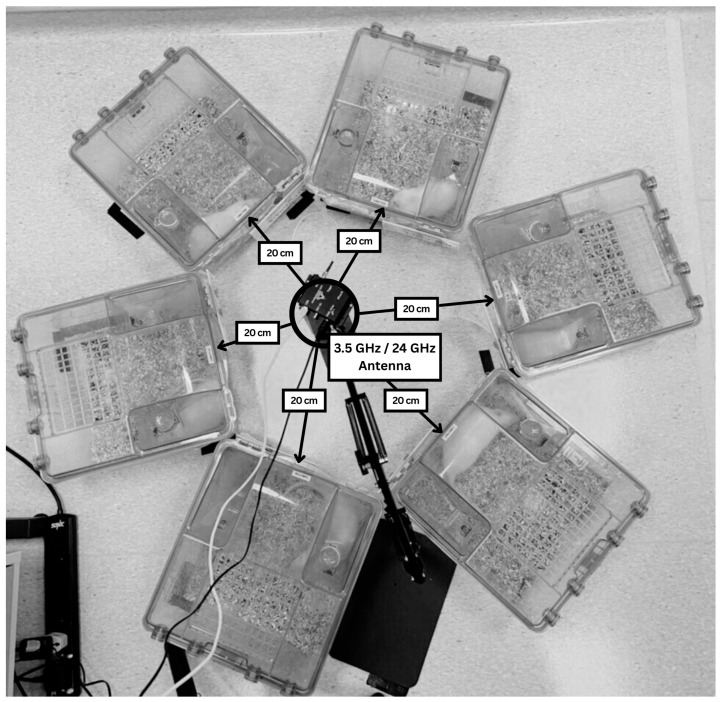
RF-EMF exposure experimental setup: single 3.5 or 24 GHz antenna at center; six cages positioned uniformly at 20 cm.

**Figure 2 biomedicines-13-02471-f002:**
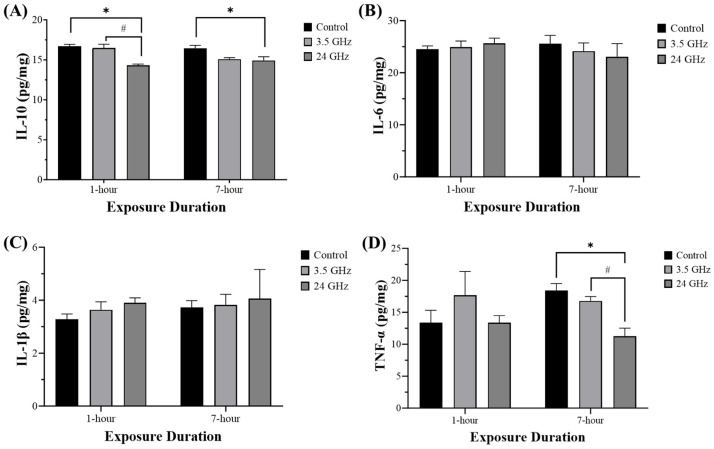
Testicular cytokines after 60-day RF-EMF exposure at 3.5 or 24 GHz for 1 h/day or 7 h/day. Bars show mean ± SEM ((**A**–**C**) n = 5–6; (**D**) n = 4–6). IL-10 was lower in the 24 GHz group versus control and 3.5 GHz at 1 h, and versus control at 7 h. TNF-α was lower in the 24 GHz group at 7 h. IL-6 and IL-1β showed no significant differences at either duration. Significance symbols: * *p* < 0.05 vs. control (same duration); # *p* < 0.05 vs. other frequency (same duration). Full ANOVA and Tukey results are provided in Appendix A.

**Figure 3 biomedicines-13-02471-f003:**
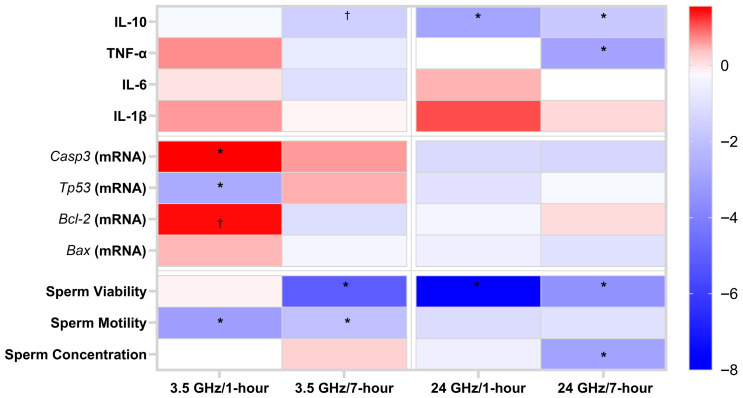
Frequency vs. duration summary heat map of signed effect sizes (Cohen’s d) for each exposure versus duration-matched control. Columns are grouped by frequency (3.5 GHz: 1 h → 7 h; 24 GHz: 1 h → 7 h). Diverging scale centered at 0 (blue = decrease, red = increase). The map highlights IL-10 suppression at 24 GHz, motility loss at 3.5 GHz, strong viability decreases (24 GHz at 1 h; both frequencies at 7 h), a 24 GHz-specific TNF-α decrease at 7 h, and transient 1-h shifts in *Tp53*/*Bcl2* that normalize by 7 h. * denotes *p* < 0.05; † denotes 0.05 ≤ *p* < 0.10. Full statistics are in Appendix A.

**Figure 4 biomedicines-13-02471-f004:**
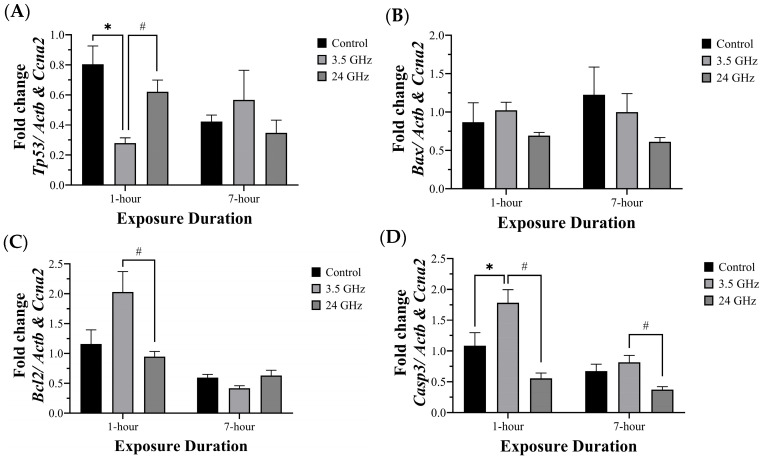
Apoptosis-related mRNA in rat testes after 60 days of RF-EMF at 3.5 or 24 GHz for 1 h/day or 7 h/day. Bars show mean ± SEM (n = 5–6). (**A**) *Tp53*: The 3.5 GHz group was lower compared with the control group at 1 h. The 24 GHz group was higher than the 3.5 GHz group at 1 h. No differences were detected at 7 h. (**B**) *Bax*: no differences at either duration. (**C**) *Bcl2*: higher in the 3.5 GHz group compared with the 24 GHz group at 1 h. No differences were detected at 7 h. (**D**) *Casp3*: higher in the 3.5 GHz group compared with the control group at 1 h and higher than the 24 GHz group at both durations. Significance symbols: * *p* < 0.05 vs. control (same duration) and # *p* < 0.05 vs. other frequency group (same duration). Full ANOVA and Tukey results are in Appendix A.

**Figure 5 biomedicines-13-02471-f005:**
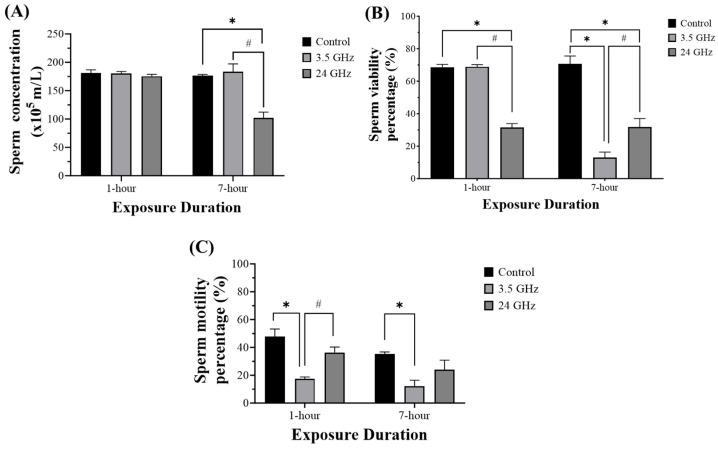
Sperm quality from the cauda epididymis after 60 days of RF-EMF at 3.5 or 24 GHz for 1 h/day or 7 h/day. Bars show mean ± SEM (n = 6). (**A**) Concentration: The 24 GHz group was lower compared with the control group and the 3.5 GHz group at 7 h. (**B**) Viability: The 24 GHz group was lower compared with the control group and the 3.5 GHz group at 1 h. At 7 h, viability was lower in both the 3.5 GHz and 24 GHz groups compared with the control group. Viability in the 24 GHz group was higher than in the 3.5 GHz group. (**C**) Motility: The 3.5 GHz group was lower compared with control at both durations and was also lower compared with the 24 GHz group at 1 h. Significance symbols: * *p* < 0.05 vs. control (same duration) and # *p* < 0.05 vs. other frequency group (same duration). Full ANOVA and Tukey results are in Appendix A.

**Table 1 biomedicines-13-02471-t001:** Genes and primer sequences used for quantitative PCR analysis of apoptosis markers in rat testis tissue.

Gene	RefSeq	Forward Primer	Reverse Primer
*Casp3*	NM_012922	5′-CCGACATCTGTGTGACTTCAC-3′	5′-CGTACAGTTTCAGCACATGG-3′
*Tp53*	NM_030989	5′-TCGTAGGTAGGCAGCTACATG-3′	5′-CGATGTCTCATCCGACTGTG-3′
*Bax*	NM_017059	5′-CGCGGTTGCTGTTGAT-3′	5′-AAGGCTCAGGCCCATCTTCT-3′
*Bcl2*	NM_016993	5′-GATGACTGAGTACCTGAACC-3′	5′-CCAGGAGAATCAAAAGGGT-3′
*Actb*	NM_031144	5′-TGCATAGGCAATGAGCGG-3′	5′-GGCATAGGCTTCTTACGGA-3′
*Ccna2*	NM_053702	5′-AGGGAAATGGAGGTTAAATG-3′	5′-CTATCAATGTAGTTCACAGCC-3′

## Data Availability

The datasets generated and analyzed during the current study are available from the corresponding author upon reasonable request.

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
