# Peer review of "Prolonged 3.5 GHz and 24 GHz RF-EMF Exposure Alters Testicular Immune Balance, Apoptotic Gene Expression, and Sperm Function in Rats"

_biomedicines, 2025, doi:10.3390/biomedicines13102471_

Round 1
Reviewer 1 Report
Comments and Suggestions for Authors
The manuscript addresses an important and timely topic, the biological impact of prolonged 5G-related RF-EMF exposure on male reproductive health. The study is well-organized, includes both mid-band (3.5 GHz) and high-band (24 GHz) frequencies, and evaluates multiple endpoints (cytokines, apoptosis markers, sperm function). The results are relevant and novel, particularly given the scarcity of data on high-frequency (mmWave) exposures. However, in order to improve the manuscript, there are some methodological and interpretative issues that need clarification before the paper can be accepted.
- First of all, Sample Size and Power. Each group included only 6 rats, which may limit statistical power. Some non-significant results (e.g., IL-1β changes) may reflect type II error. A post-hoc power analysis should be provided, or this limitation explicitly stated.
-
Statistical issues: While ANOVA results are reported with F, p, and R², the presentation is heavy and could be simplified (e.g., summarizing in supplementary tables). Effect sizes (η² or Cohen’s d) would help quantify biological relevance.
-
Mechanicistic and Interpretations issuses: The discussion speculates on ROS-mediated pathways but no oxidative stress markers (MDA, SOD, CAT, GPx) were measured. This is a significant gap. Please make it clearer that ROS involvement is hypothetical in this study. Similarly, no hormonal assays (testosterone, LH, FSH) were performed, despite endocrine disruption being mentioned in the discussion. Consider acknowledging this limitation. Moreover, the data show IL-10 and TNF-α suppression at 24 GHz, but IL-6 and IL-1β remained unchanged in most cases. The interpretation as a “non-classical immune profile” is interesting but requires a clearer biological explanation. Could systemic stress responses or altered Sertoli cell signaling explain this selective modulation?
-
Exposure System and Dosimetry mhetod: exposure setup is described, but the lack of measured SAR values and thermal monitoring is a major limitation. The paper acknowledges this, but it should be emphasized more strongly in both Methods and Discussion. Field uniformity and actual exposure levels were validated in situ (not just via CST simulations). Without this, biological effects cannot be confidently attributed to RF-EMF.
- The exposure system does not reproduce real 5G modulation or beamforming. This should be discussed in more depth. Otherwise, readers may overinterpret the findings as directly applicable to human exposures.
Minor revisions:
-
Figures are clear but legends are very dense. Consider simplifying and moving ANOVA details to supplementary material.
-
In the Abstract, avoid speculative terms like “systemic alterations” without clear evidence.
Reviewer 2 Report
Comments and Suggestions for Authors
Comment 1
The authors claim, based on experiments in rats, that RF-EMF can affect the testes even at levels that comply with international safety guidelines based on thermal effects. However, as the authors themselves state in the limitations—“Although SAR levels were controlled according to ICNIRP guidelines, direct thermal measurements in testicular tissue were not conducted. This limits the ability to confirm whether subtle thermal elevation contributed to the observed changes in cytokine suppression, apoptotic gene expression, or sperm viability (Houston et al. 2019; Kesari et al. 2021).”—the evidence is not strong enough to conclude that the observed effects represent genuine non-thermal biological effects. This leaves readers with a sense of partial inconsistency, which is unfortunate. Furthermore, since ICNIRP and similar safety guidelines are intended for human protection, it is questionable whether they can be directly applied to rats, whose bodies are much smaller.
An ideal way to resolve this issue would be to design a biological model in which the theoretical energy delivered by RF-EMF is reproduced through thermal exposure delivered by non-RF methods. This would allow one to distinguish whether the observed testicular effects are truly non-thermal biological effects, or simply a reflection of subtle thermal changes.
If such additional experiments cannot be performed, the authors should either (a) cite prior studies that support the interpretation of the observed changes as non-thermal effects, or (b) soften their current claims about non-thermal biological effects.
Comment 2
In this study, RF‑EMF exposure was delivered at 22 dBm for the 3.5 GHz group and 11 dBm for the 24 GHz group, from an antenna positioned 20 cm above the cages for either 1 h or 7 h per day over 60 days. The authors note that preliminary simulations estimated SAR values within ICNIRP/IEEE guidelines. Given the distance and radiated power, the resulting power densities would be orders of magnitude below the public exposure limits. Readers from reproductive medicine or public health backgrounds are not necessarily versed in RF engineering; it would therefore be helpful to include a more explicit discussion of the expected power density at the animal’s surface and how this compares with recognised safety limits.
Comment 3
This study does not claim that “rats exposed to RF-EMF at these intensities will experience the same outcomes as humans.” Rather, it suggests that “even weak RF-EMF exposures, which should not produce thermal effects, can still affect reproductive function in rats; therefore, it is worth reconsidering whether current human safety standards are truly adequate.” I believe this is the correct interpretation of the authors’ message. However, I needed to read the manuscript several times before arriving at this interpretation. Perhaps my difficulty was due to my own limitations as a reader, but it raises the question of whether the message could be presented more clearly.
For example, the Conclusion section is quite long, and this may reduce clarity. Generally, one paper should have a single, concise take-home message. Condensing the Conclusion section to focus on the essential point would improve readability. In addition, because the topic of electromagnetic fields is sometimes exploited by conspiracy theorists for unscientific arguments, it is important to present the claims in a particularly careful and balanced manner.
Another consideration is that in this experiment, rats were positioned so that their entire bodies were only tens of centimeters away from the antenna, resulting in whole-body RF absorption. In contrast, human exposure from 5G base stations or mobile phones typically involves localized surface exposure to small areas of the body. This is especially true for 24 GHz millimeter waves, which penetrate only a few millimeters into the skin. Human testes, being located deep inside the body, are therefore unlikely to be directly reached by 24 GHz radiation.
Additionally, in this experiment, exposures consisted of continuous near-sinusoidal waves applied for 1 or 7 hours per day. In reality, 5G communication involves beamforming, modulation, and intermittent transmission, producing exposure patterns very different from those in this study. These differences make it difficult to directly extrapolate the findings to human 5G usage. For this reason, the manuscript should be careful to avoid excessive generalization that might unnecessarily alarm the general public.
Comment 4
The differences in biological effects between 3.5 GHz and 24 GHz are not clearly conveyed. From the data, it appears that 24 GHz leads to more pronounced suppression of sperm viability and anti‑inflammatory signalling during long‑term exposure, whereas 3.5 GHz mainly reduces motility and increases caspase‑3 activity. If this is indeed the authors’ interpretation, please illustrate it with an additional figure or graphical abstract. The transient changes in Tp53 and Bcl‑2 expression—significant after 1 h but not after 7 h—also deserve visualisation, as they suggest early gene responses that normalise over time. Likewise, it might be helpful to visualise the lack of significant changes in IL‑6, IL‑1β and TNF‑α to emphasise that classical pro‑inflammatory pathways were not activated.
Comment 5
For Bcl‑2 at 1 h and Casp3 at 7 h, Tukey’s post‑hoc test found a significant difference only between the 3.5 GHz and 24 GHz groups, whereas neither group differed significantly from the control. This pattern implies that the two frequencies affected the endpoints in opposite directions or to different extents. I would appreciate the authors’ interpretation of these findings.
Comment 6
In Figures 1D, 2A, 2C and 3C, the means for the two control groups (1 h and 7 h) appear different, yet no statistical comparison between the control groups is reported. Since these are independent groups of animals, I assume the differences reflect random biological variation; however, clarification from the authors would be welcome.
Comment 7
While the study measured cytokines, apoptosis‑related genes and sperm parameters, no histopathological examination of the testes or epididymides was performed. Histology (e.g. Johnsen’s scoring) would allow direct assessment of Sertoli cell or seminiferous tubule structure and would strengthen the toxicological evaluation. Please consider including histological analyses or explaining why they were omitted.
Comment 8
The WHO 2010 guidelines for semen analysis classify sperm motility as progressive, non‑progressive and immotile; these criteria were developed for human semen. Because rat sperm differ in morphology and motility patterns, species‑specific motility assessments would be preferable. Please clarify your rationale for applying the WHO human motility classification to rat sperm and discuss any limitations this may introduce.
Round 2
Reviewer 1 Report
Comments and Suggestions for Authors
The suggested implementations have been done point by point, I have no further comments.
Author Response
Comment 1 :
The suggested implementations have been done point by point, I have no further comments.
Response 1:
We appreciate the reviewer’s careful reading and confirmation that our point-by-point revisions are satisfactory. Thank you for your time and constructive input throughout the process.
Reviewer 2 Report
Comments and Suggestions for Authors
Thank you for the authors’ thorough revisions. These changes have resolved my previous concerns. I have one remaining suggestion related to Comment 4 from the first review.
While your revisions have clarified the frequency-specific findings, I believe the manuscript could be further strengthened by including an additional figure or a graphical abstract. According to Biomedicines’ Instructions for Authors, “A graphical abstract (GA) is an image that appears beneath the text abstract. In addition to summarizing the content, it should represent the topic of the article in an attention‑grabbing way.” I encourage the authors to prepare a GA that succinctly visualizes the study’s objectives and key findings. Please ensure, as noted previously, that the manuscript avoids any over‑generalization that might unnecessarily alarm the general public.
In addition, it would be helpful to provide a photograph of the actual RF‑EMF exposure setup used for the rats in Section 2.1.2 “Exposure Experimental Setup.” Including such an image would improve reproducibility and assist readers in understanding how the exposures were carried out.
Author Response
Comment 1:
Thank you for the authors’ thorough revisions. These changes have resolved my previous concerns. I have one remaining suggestion related to Comment 4 from the first review.
While your revisions have clarified the frequency-specific findings, I believe the manuscript could be further strengthened by including an additional figure or a graphical abstract. According to Biomedicines’ Instructions for Authors, “A graphical abstract (GA) is an image that appears beneath the text abstract. In addition to summarizing the content, it should represent the topic of the article in an attention‑grabbing way.” I encourage the authors to prepare a GA that succinctly visualizes the study’s objectives and key findings. Please ensure, as noted previously, that the manuscript avoids any over‑generalization that might unnecessarily alarm the general public.
In addition, it would be helpful to provide a photograph of the actual RF‑EMF exposure setup used for the rats in Section 2.1.2 “Exposure Experimental Setup.” Including such an image would improve reproducibility and assist readers in understanding how the exposures were carried out.
Response 1:
Thank you for this helpful suggestion. We agree and have implemented both additions in line with Biomedicines’ guidance and our prior commitment to avoid over-generalization.
Locations of revisions in the manuscript
-
Page 2: Graphical Abstract inserted beneath the text abstract
-
Page 5: Section 2.1.2 Figure 1 (Animal exposure experimental setup) inserted.
Round 3
Reviewer 2 Report
Comments and Suggestions for Authors
The authors appear to have fully implemented all requested revisions. I commend the authors for their thorough and careful work.
Minor editorial adjustments in accordance with the Instructions for Authors can be entrusted to the journal’s excellent editorial staff.